# Targeting Inflammation and Iron Deficiency in Heart Failure: A Focus on Older Adults

**DOI:** 10.3390/biomedicines13020462

**Published:** 2025-02-13

**Authors:** Daniela Maidana, Andrea Arroyo-Álvarez, Guillermo Barreres-Martín, Andrea Arenas-Loriente, Pedro Cepas-Guillen, Raphaela Tereza Brigolin Garofo, Pedro Caravaca-Pérez, Clara Bonanad

**Affiliations:** 1INCLIVA—Biomedical Research Institute, 46010 Valencia, Spain; danumaidana.dm@gmail.com (D.M.); andreastream97@gmail.com (A.A.-Á.);; 2Cardiology Department, Hospital Clínic Barcelona, 08036 Barcelona, Spainpecarav86@gmail.com (P.C.-P.); 3Quebec Heart & Lung Institute, Laval University, Quebec City, QC G1V 4G5, Canada; pedro.cepasguillen@gmail.com; 4Cardiology Department, Hospital do Coração, São Paulo 04004-030, Brazil; raphaelagarofo@outlook.com; 5Cardiology Department, University Clinical Hospital of Valencia, 46010 Valencia, Spain

**Keywords:** heart failure, iron deficiency, inflammation, older adults

## Abstract

**Background/Objectives**: Heart failure (HF) is a leading cause of morbidity and mortality worldwide, with a higher prevalence among older adults. Iron deficiency (ID), affecting up to 50% of HF patients, is closely linked to chronic inflammation, exacerbating HF outcomes. This review aims to explore the interplay between inflammation, ID, and HF, focusing on older patients, and to identify therapeutic gaps and emerging treatment strategies. **Methods**: A comprehensive review of the literature was conducted, emphasizing the pathophysiological mechanisms of inflammation and ID in HF, the challenges of current diagnostic criteria, and the limitations of available treatments. Emerging pharmacological and diagnostic approaches were analyzed. **Results**: Chronic inflammation in HF, particularly in older adults, promotes functional ID through elevated hepcidin levels, impairing iron availability and worsening anemia. Current diagnostic criteria, relying heavily on ferritin, often misclassify ID due to inflammation. Intravenous (IV) iron therapy shows clinical benefits in patients with <50% left ventricular ejection fraction (LVEF), but the evidence is limited in heart failure with preserved ejection fraction (HFpEF). Emerging therapies, such as Sodium-Glucose Cotransporter-2 inhibitors (SGLT2is) and prolyl hydroxylase inhibitors like Roxadustat, offer promising avenues to improve iron metabolism and outcomes. **Conclusions**: ID and inflammation significantly impact HF progression, particularly inolder adults. Refining diagnostic criteria and exploring innovative therapies are critical to addressing these challenges. Future research should prioritize personalized approaches targeting inflammation and ID, especially in underrepresented populations, such as HFpEF and elderly patients.

## 1. Introduction

Heart failure (HF) remains a leading cause of morbidity and mortality worldwide [1]. The prevalence of HF ranges from 1% to 5%, with significantly higher rates among older adults, and is expected to rise further, driven in part by an aging population [2]. However, these numbers may be underestimated, as HF is often underdiagnosed in older patients due to its overlap with geriatric syndromes such as frailty and multimorbidity [3], reinforcing the importance of addressing HF in these patients. These factors contribute to an increase in the absolute number of HF hospitalizations, projected to grow by as much as 50% over the next 20 years [4], posing a substantial public health challenge.

In older adults, several aging-related factors, along with traditional risk factors such as hypertension, diabetes, and coronary artery disease, trigger structural and functional changes in the heart, increasing the incidence of HF [5]. One frequently overlooked but crucial aspect is iron deficiency (ID), which affects up to 50% of HF patients [6]. ID is closely linked to chronic low-grade inflammation, a condition commonly associated with aging [7]. The concept of “inflammaging” highlights how persistent inflammatory pathways in older adults contribute to an increased cardiovascular risk. Consequently, aging, chronic inflammation, and ID collectively exacerbate HF [8].

ID has emerged as a significant factor in heart failure with reduced ejection fraction (HFrEF) or heart failure with mildly reduced ejection fraction (HFmrEF), associated with a worse quality of life and a higher risk of adverse events. ID treatment has been shown to improve exercise tolerance and quality of life and reduce hospitalizations and mortality [4]. However, further evidence is needed on managing ID, especially in patients with heart failure with preserved ejection fraction (HFpEF) and older adults [9]. The divergent results from trials evaluating the role of ID in HF underscore the need for further research into the pathophysiology of HF and highlight the necessity of refining current definitions of ID in HF to improve patient outcomes [10,11].

This review aims to provide a comprehensive overview of the main pathophysiological mechanisms driving inflammation and ID in HF, focusing on older patients. We will address the ongoing controversies, the limitations of current treatments, and gaps in our understanding while discussing the latest pharmacological advancements and pivotal studies in this evolving field to improve patient outcomes.

## 2. Methods

A detailed literature review was conducted to comprehensively understand the interplay between heart failure, inflammation, and iron deficiency. The search included PubMed, Scopus, and Web of Science databases, focusing on articles published between 2010 and 2025, particularly in the last five years. Studies were selected based on their relevance to the relationship between these factors, especially in older adults. Keywords such as ‘heart failure’. ‘inflammation’, ‘iron deficiency’, and ‘older adults’ guided the search to ensure a thorough and focused analysis.

### Scope of the Review

This review addresses the ongoing controversies, the limitations of current treatments, and gaps in our understanding while discussing the latest pharmacological advancements and pivotal studies in this evolving field to improve patient outcomes.

## 3. Pathophysiology of Heart Failure: Role Inflammation and Iron Deficiency

HF represents the final common pathway of numerous cardiovascular pathologies, driven by a complex interplay of hemodynamic, neurohormonal, metabolic, and inflammatory factors contributing to its onset and progression. Inflammation plays a pivotal role in the pathophysiology of HF by regulating key processes like fibrosis and ventricular remodeling [12]. In older patients, chronic low-grade inflammation and immunosenescence, combined with comorbidities, accelerate cardiac dysfunction, a progression that is particularly notable in those patients with ID [13]. In this section, we discuss how inflammation is linked with HF pathophysiology, the role of “inflammaging” in older adults, and the relationship of these processes to ID in this context (Figure 1).

### 3.1. Inflammation in Heart Failure

Inflammation in HF can be categorized into acute or chronic [12]. Acute inflammatory episodes, such as those following an acute myocardial infarction (AMI), trigger an intense inflammatory response aimed at repairing tissue damage [14]. However, when this response is prolonged, it contributes to adverse myocardial remodeling, ultimately leading to HF [15]. During AMI, the massive release of proinflammatory cytokines like Tumor Necrosis Factor-alpha (TNF-α) and Interleukin-6 (IL-6) promotes the infiltration of inflammatory cells, which can lead to damage and the loss of cardiomyocytes, as well as fibrosis [16].

Chronic low-grade inflammation is even more insidious, occurring in individuals with comorbidities and closely associated with aging. These conditions create an unresolved inflammatory state involving effectors such as proinflammatory cytokines and components of the innate and humoral immune response, which contribute to the progressive dysfunction of the left ventricle (LV) [17]. Both HFrEF and HFpEF are influenced by inflammation but with distinct mechanisms and outcomes [11].

In HFrEF, the inflammatory process primarily targets cardiomyocytes, leading to their apoptosis and necrosis, which drives the eccentric remodeling of the LV, resulting in ventricular dilation and decreased contractile function [18]. In contrast, HFpEF involves endothelial dysfunction, increasing myocardial stiffness through interstitial fibrosis and impairing diastolic function [18]. HFpEF is now recognized as a systemic syndrome, where inflammation, aging, lifestyle factors, genetic predisposition, and comorbidities such as age, diabetes, hypertension, obesity, and atrial fibrillation play a significant role [19]. All of these factors lead to fibrosis, nitric oxide signaling deficits, and mitochondrial dysfunction, contributing to the structural and functional changes in the heart [20]. Finally, all of these inflammatory processes, affecting cardiomyocytes in HFrEF or endothelial function in HFpEF, are associated with molecular patterns that drive immune responses and contribute to disease progression [21].

The interaction between pathogens, tissue damage, and immune responses in HF can be understood through the concepts of pathogen-associated molecular patterns (PAMPs), damage-associated molecular patterns (DAMPs), and lifestyle-associated molecular patterns (LAMPs) [22]. PAMPs originate from pathogens, and DAMPs from damaged tissues like after an AMI, while LAMPs are influenced by lifestyle factors like poor diet or inactivity. All of these patterns can dysregulate the immune system, leading to unresolved inflammation, chronic damage, and worsening HF [22]. Table 1 summarizes the main aspects that differentiate inflammation in HF.

### 3.2. Inflammation in the Older Patient

In older patients, the phenomenon of inflammaging—a chronic, low-grade inflammation that accompanies aging—plays a central role in the progression of HF and the development of ID [23].

Inflammaging involves chronic innate immune activation, leading to the release of inflammatory cytokines as Interleukin-1 (IL-1), TNF-α, and IL-6 that exacerbate HF mechanisms like Nuclear factor kappa B (NF-κB) activation and NOD-like receptor family pyrin domain containing 3 (NLRP3) inflammasome formation drive systemic inflammation, oxidative stress, arterial stiffness, and endothelial dysfunction, increasing cardiovascular risk [24]. This creates a cycle of inflammation that accelerates HF progression and adverse outcomes [25]. The upregulation of proinflammatory mediators like Interleukin-1β (IL-1β) and Interleukin-18 (IL-18) further exacerbates this, as both are implicated in cardiac fibrosis, hypertrophy, and diastolic dysfunction [26].

Moreover, in patients with HF, elevated levels of reactive oxygen species (ROS) damage cardiomyocytes, while chronic inflammation impairs iron regulation, worsening ID and its impact on HF [27]. In addition, the infiltration of proinflammatory M1 macrophages into the myocardium was observed more prominently in aging women, and reduced the presence of anti-inflammatory M2 macrophages and contributed to cardiac dysfunction. In contrast, male patients with end-stage cardiomyopathy display a higher presence of CD68-positive macrophages, highlighting sex differences in immune responses associated with inflammaging and HF progression [24].

### 3.3. The Role of Inflammation in Iron Deficiency

The link between inflammation and ID in HF is well-established and especially relevant in older patients [28]. Under conditions of chronic inflammation, as seen in HF and inflammaging, levels of IL-6 increase, which triggers hepcidin production [29]. Hepcidin reduces iron absorption in the intestines and sequesters iron in macrophages and hepatocytes. This primarily leads to functional ID, where iron stores are adequate but unavailable for essential processes such as hemoglobin synthesis and adenosine triphosphate (ATP) production in mitochondria [30]. Several studies have demonstrated the relationship between inflammation and ID [31,32,33]. Specifically, an international registry including 2329 HF patients found that increased plasma IL-6 concentrations were associated with lower iron, highlighting IL-6’s role in the dysregulation of iron metabolism [29]. Inflammatory cytokines also inhibit erythropoiesis by impairing the bone marrow’s response to erythropoietin, contributing to anemia [34]. In patients with HF, exacerbated inflammation promotes this state of functional ID. This interaction between inflammation, ID, and anemia increases stress on the heart and exacerbates HF symptoms, including fatigue and exercise intolerance [35].

### 3.4. Iron Deficiency in Heart Failure

ID is standard in patients with HF, affecting up to 50% of them, regardless of the presence of anemia [36]. According to HF practice guidelines, ID in HF is classified into two primary forms: absolute and functional. Absolute ID occurs when total body iron stores are depleted, while functional ID occurs when iron is sequestered in the body despite adequate stores, usually due to inflammation [27]. Both forms of ID are associated with poorer outcomes in HF patients, including reduced exercise tolerance, higher hospitalization rates, and increased mortality [37].

As previously explained, ID impacts hematopoietic function and directly affects cellular function due to its role in mitochondrial respiration and energy production [27]. In HF patients, ID is associated with reduced functional capacity, decreased quality of life, and increased hospitalizations and mortality [38]. In HFrEF, ID directly contributes to myocardial dysfunction by reducing the heart’s energy consumption. But ID not only impacts HFrEF, it is also prevalent in HFpEF, though the mechanisms by which it impacts cardiac and muscular performance may differ slightly from those observed in HFrEF [39]. Rather than contractility failure, HFpEF patients experience ventricular stiffness and diastolic dysfunction exacerbated by inflammation and ID [40].

## 4. Impact of Iron Deficiency on Heart Failure in Older Patients

In older adults with HF, a vicious cycle emerges from the interplay of multimorbidity, inflammaging, and inflammation [41]. Chronic inflammation, often driven by comorbidities, elevates hepcidin levels, leading to functional ID. This interaction significantly worsens prognosis, increasing hospitalizations and mortality [4]. The intricate relationship between aging, inflammation, and ID in HF highlights the need for a deeper understanding of these mechanisms to improve patient outcomes.

### The Role of Aging and Inflammation in Iron Deficiency

Aging-related changes in immune function, termed immunosenescence and inflammaging, synergistically worsen iron dysregulation in older HF patients [23]. In this population, the inflammatory state becomes even more pronounced, as HF is associated with elevated levels of proinflammatory cytokines.

In this context, hepcidin, a central iron homeostasis regulator, leads to functional ID [42]. As a result, HF patients with ID experience disrupted mitochondrial function, impairing the electron transport chain and reducing oxidative phosphorylation, forcing the heart to rely on glycolysis for energy production [27]. This shift is partly driven by the Hypoxia-Inducible Factor (HIF) pathway, as ID stabilizes HIF by inhibiting prolyl hydroxylases, even in normoxic conditions. The activation of HIF target genes mimics a hypoxic response, further promoting glycolysis over oxidative metabolism. While glycolysis generates energy faster, it is less efficient, leading to diminished ATP production and compromised cardiac contractility, worsening HF outcomes [35].

Furthermore, the combined effects of aging, inflammation, and ID severely impair cardiac function and elevate the risk of hospitalizations in older HF patients [38]. Figure 2 summarizes how the combined impact of inflammation from HF and aging exacerbates ID, leading to increased hospitalizations and a worse overall prognosis.

A systematic review and meta-analysis examined the relationship between anemia and frailty in older adults [43]. The study found that frail individuals are more than twice as likely to present with anemia as their non-frail counterparts, underscoring the close association between these two conditions. The pooled prevalence of frailty among individuals with anemia was 24%, further highlighting the need to consider frailty when assessing OA with anemia [43].

On the other hand, a study involving 208 older hospitalized patients categorized by frailty revealed that inflammation significantly increased the prevalence of anemia, particularly in frail patients [44].

The diagnosis of ID in older patients is essential for improving outcomes. In older adults with multimorbidity, factors like frailty, malnutrition, and chronic diseases often contribute to anemia, making it challenging to differentiate ID related to HF from other causes of anemia. This challenge is especially relevant in frail patients, who are more likely to have elevated markers of inflammation that can obscure ID [45].

## 5. Diagnosis, Treatment, and Therapeutic Challenges

Early diagnosis is critical because treating ID has been shown to improve clinical outcomes, reduce hospitalizations, and enhance the quality of life [4]. A retrospective study of older patients with HF showed that addressing ID early, particularly with intravenous (IV) iron, reduced the incidence of infections and improved functional capacity [46]. Therefore, screening for ID in older patients with HF should be a routine part of clinical care, as these patients are at a higher risk of complications from untreated ID.

The diagnosis of ID in HF patients relies on serum ferritin and transferrin saturation (TSAT) as primary biomarkers. Current guidelines define ID as a ferritin level <100 ng/mL or 100–299 ng/mL with TSAT < 20% [4]. However, this definition has sparked significant controversy due to the distortion of ferritin levels by systemic inflammatory states, commonly seen in older adults and patients with chronic conditions like inflammatory bowel disease or chronic kidney disease (CKD) [31]. Additionally, as ferritin is an acute-phase reactant, its levels are often elevated in response to inflammation, potentially masking actual ID [47].

Packer et al. highlighted that nearly 25 years ago, the diagnostic threshold for ferritin in patients with HF was significantly increased—by 5- to 20-fold—to promote the use of iron supplements and improve the effectiveness of erythropoiesis-stimulating agents in treating renal anemia [10]. However, this change was driven by clinical necessity rather than studies on total body iron depletion. As a result, this threshold is not a reliable ID indicator in HF patients. He proposes that, in the context of HF, a TSAT threshold of less than 20% should be prioritized over ferritin levels to diagnose ID [10]. This approach emphasizes the need for a more precise ID marker, which helps identify patients who would most benefit from IV iron therapy [10].

Shifting the focus from ferritin to TSAT < 20% could lead to earlier interventions and better outcomes for HF patients, particularly older adults prone to both inflammation and ID [10,28]. This change in diagnostic criteria is particularly relevant in light of the evidence that patients with functional ID (low TSAT but normal ferritin) respond well to IV iron supplementation [10].

### 5.1. Clinical Trials Supporting Intravenous Iron Therapy

Current guidelines recommend IV iron therapy with ferric derisomaltose or ferric carboxymaltose for patients with HF and <50% left ventricular ejection fraction (LVEF) and ID, supported by evidence from clinical trials outlined in Table 2 [4]. These studies consistently show that IV iron alleviates symptoms, enhances functional capacity, and reduces hospitalizations in this population. However, despite the prevalence of ID and associated symptoms, similar recommendations do not extend to patients with HFpEF, where the evidence is less [40].

Regarding the differences between the use of one type of IV iron versus another, the IRONMAN and AFFIRM-AHF studies evaluated the administration of IV iron in patients with HF and ID, showing similar benefits in reducing hospitalizations. However, no statistical significance was reached in the reduction of cardiovascular mortality. Both studies suggest that IV iron, regardless of the type of complex used, provides a general benefit in these patients.

On the other hand, a study that evaluated the cost-effectiveness of ferric derisomaltose versus ferric carboxymaltose in patients with inflammatory bowel disease and ID anemia found that ferric derisomaltose not only improved patient quality of life but also reduced direct healthcare costs compared to ferric carboxymaltose [48].

Additionally, a systematic review and meta-analysis, which pooled data from four large-scale randomized controlled trials, suggested that ferric isomaltose was associated with a lower incidence of cardiovascular adverse events compared to ferric carboxymaltose in patients with ID anemia of various etiologies [49].

Potential explanations for these differences include the depletion of adenosine triphosphate and 2,3-diphosphoglycerate (critical intermediate metabolites) arising from hypophosphatemia, specifically proposed as a possible mechanism for cardiomyopathy and arrhythmia. It has been observed that ferric carboxymaltose results in a higher incidence of post-infusion hypophosphatemia compared to ferric derisomaltose [50,51]. However, further research is needed to confirm whether this also influences cardiovascular events. Recent studies suggest that ferric isomaltose may be associated with a lower incidence of severe hypersensitivity reactions than ferric carboxymaltose [52].

**Table 2 biomedicines-13-00462-t002:** Key clinical trials supporting intravenous iron therapy in heart failure. Evidence from “2023 Focused Update of the 2021 ESC Guidelines for diagnosing and treating acute and chronic heart failure”. ID: iron deficiency; LVEF: left ventricular ejection fraction; T: TSAT, transferrin saturation (%); F: ferritin (µg/L); CVM: cardiovascular mortality; HHF: hospitalization for heart failure; IV: intravenous.

Study	Number of Patients (n)	Patient Population and ID Definition	Intervention	Primary Outcomes	Key Findings
IRONMAN (2022) [53]	1137	HF patients (LVEF ≤ 45%). TSAT < 20% + ferritin < 400 or ferritin < 100.	Ferric derisomaltose vs. usual care	Composite of total HF hospitalizations and cardiovascular mortality (CVM).	RR 0.82 (95% CI 0.66–1.02; *p* = 0.070); significant benefit post-COVID-19 analysis (HR 0.76, *p* = 0.047) in reducing primary endpoint.
Graham et al. (2023) [54]	3373	Meta-analysis of 10 trials; ferritin < 100 or ferritin 100–300 + TSAT < 20%. IRONMAN also allowed TSAT < 20% with ferritin < 400.	IV iron therapy vs. standard care	Composite of total hospitalization for heart failure (HHF) and CVM.	IV iron reduced total HHF and CVM (RR 0.75, 95% CI 0.61–0.93; *p* < 0.01), and first HF hospitalization or cardiovascular death (OR 0.72, *p* = 0.04).
AFFIRM-AHF (2020) [55]	1108	HF patients (LVEF < 50%). Ferritin < 100 or TSAT < 20% + ferritin 100–299.	Ferric carboxymaltose vs. placebo	Total HF hospitalizations and CVM.	Ferric carboxymaltose reduced the risk of first HF hospitalization (RR 0.74; 95% CI 0.58–0.94; *p* = 0.013) and recurrent hospitalizations.
PIVOTAL (2019) [56]	2141	End-stage CKD patients on hemodialysis, with ferritin < 400 and TSAT < 30%, and receiving erythropoiesis-stimulating agents.	High-dose IV iron vs. low-dose iron	First and recurrent HF events.	High-dose IV iron reduced HHF (HR: 0.66; 95% CI: 0.48–0.91; *p* = 0.01) in hemodialysis patients.
CONFIRM-HF (2015) [57]	301	HF patients (LVEF ≤45%) with ferritin < 100 or TSAT < 20% + ferritin 100–300.	Ferric carboxymaltose vs. placebo	Change in 6-min walk distance (6MWD) at 24 weeks.	Significant improvement in 6MWD (+33 ± 11 m, *p* = 0.002); reduced HHF. HR 0.39; 95% CI: 0.19–0.82; *p* = 0.009.

### 5.2. Limitations and Controversies in Iron Supplementation

Although trials like FAIR-HF, CONFIRM-HF, AFFIRM-AHF, and EFFECT-HF (as outlined in Table 2) demonstrated the benefits of IV ferric carboxymaltose in improving symptoms, exercise capacity, and reducing hospitalizations in patients with HF and ID, recent findings from the HEART-FID trial have raised new questions regarding its impact on hard clinical endpoints [58].

In HEART-FID, IV ferric derisomaltose was associated with numerical improvements in all-cause mortality, heart failure hospitalizations, and 6 min walk distance. However, these differences did not reach statistical significance compared to placebo, likely due to the trial’s rigorous methodological design [58]. Unlike previous studies, HEART-FID employed strict inclusion criteria, enrolling a large and heterogeneous population with detailed stratification by comorbidities and baseline characteristics. Additionally, the study featured an extended follow-up period and comprehensive assessments of patient outcomes, reducing the potential for bias but making it more challenging to demonstrate significant differences in mortality and hospitalizations [58].

These findings highlight the complexity of evaluating iron supplementation in HF populations and suggest that methodological rigor may influence outcome detectability. Moreover, differences in trial design, patient selection, and iron formulations (ferric carboxymaltose vs. ferric derisomaltose) could have contributed to the discrepancies observed across studies. Further research is needed to determine whether specific subgroups of HF patients, particularly those with more severe iron deficiency or a more significant inflammatory burden, may benefit most from iron supplementation. More extensive and longer-duration trials with refined patient stratification are required to fully elucidate the role of IV iron in improving hard clinical endpoints such as mortality and hospitalizations.

### 5.3. Response in Heart Failure with Preserved Ejection Fraction

While ID is standard in HFpEF patients, the current guidelines do not advocate for routine iron supplementation [4]. This discrepancy is primarily due to the limited evidence regarding the efficacy of iron treatment in this population. For example, the ferric carboxymaltose and exercise capacity in HFpEF and ID: the FAIR-HFpEF trial investigated the impact of IV iron on exercise capacity and the quality of life in patients with HFpEF and serum ferritin < 100 ng/mL or ferritin 100–299 ng/mL with TSAT < 20%. Although some patients experienced improvements, the overall results did not justify a broad recommendation of iron therapy in this group because the trial lacked sufficient power to identify or refute effects on symptoms or quality of life. Consequently, many patients with HFpEF continue to suffer debilitating symptoms, yet there is insufficient evidence to support routine iron supplementation [59].

The Trial of Exercise Training in Patients with Heart Failure and Preserved Ejection Fraction (TRAINING-HF) was a landmark study designed to evaluate the efficacy of structured exercise training in improving functional capacity and quality of life in patients with HFpEF. The trial demonstrated that tailored exercise programs significantly enhance peak oxygen consumption (peak VO2), functional status, and symptoms in HFpEF patients, providing robust evidence for the role of physical training in this population. Moreover, the findings highlighted the influence of patient-specific factors, such as ID, on the overall response to therapy [60].

Building on these findings, a sub-study of TRAINING-HF by Palau et al. provides additional insights into the specific role of ID in modulating exercise response in HFpEF. This sub-study revealed that patients with ID, defined as ferritin levels < 100 ng/mL or TSAT < 20%, exhibited a markedly poorer response to exercise interventions compared to non-ID patients (*p* for interaction < 0.001). Specifically, individuals with lower baseline ferritin or TSAT showed less improvement in peak VO2, a key marker of aerobic capacity and functional status [61]. These findings underscore the importance of evaluating and addressing iron status as part of the pre-exercise assessment in HFpEF patients to optimize therapeutic outcomes.

### 5.4. Impact of Pharmacologic Treatments on Iron Metabolism

Pharmacologic treatments for HF, such as Angiotensin-Converting Enzyme inhibitors (ACEis) and Angiotensin II Receptor Blockers (ARBs), may influence iron metabolism. ACEis and ARBs may affect iron metabolism by suppressing erythropoietin production [62]. This medication could exacerbate anemia and worsen ID in some patients [63]. Although these drugs improve cardiovascular outcomes by modulating the Renin–Angiotensin–Aldosterone System (RAAS), their impact on iron regulation needs further exploration.

Conversely, Sodium-Glucose Cotransporter-2 inhibitors (SGLT2is) have shown promise in improving iron homeostasis by reducing hepcidin levels. By lowering hepcidin, SGLT2i may enhance iron mobilization, improving ID and HF outcomes in patients across the ejection fraction spectrum [64].

Finally, ACEis, such as sacubitril/valsartan, may influence iron metabolism. These drugs lower hepcidin and ferritin levels by reducing inflammation, improving iron availability, and promoting erythropoiesis [65]. However, they may also distort the interpretation of iron biomarkers, complicating the diagnosis and management of ID in HF [10]. Similarly, vericiguat has been associated with an increased incidence of anemia, as observed in the VICTORIA trial [66]. However, the underlying pathophysiological mechanism remains unclear due to the lack of ferric metabolism biomarkers collected during the study.

### 5.5. Clinical Implications and Management

Despite compelling evidence that TSAT < 20% is a more reliable marker of ID than ferritin, current clinical guidelines have not yet fully adopted this approach [4]. It is argued that reliance on ferritin thresholds is outdated, and TSAT is proposed to be prioritized to identify better patients who would benefit from IV iron therapy. This shift would align with the growing evidence showing that patients with hypoferremia (TSAT < 20%) respond best to iron supplementation [10]. The role of pharmacologic treatments in managing ID also requires careful consideration. As noted, ACEis, ARBs, neprilysin inhibitors, and SGLT2is influence iron metabolism differently. Clinicians should consider these effects when interpreting iron biomarkers and making treatment decisions. For instance, SGLT2is may reduce hepcidin and improve iron availability, while ACEis and ARBs may exacerbate anemia by suppressing erythropoiesis.

Managing ID in HF, particularly in older adults, is essential for improving clinical outcomes. Recent guidelines emphasize the importance of regularly assessing iron status—measuring ferritin and TSAT levels—to identify and treat ID early [4]. While IV iron therapy has emerged as a valuable treatment for patients with symptomatic ID with HF and <50% LVEF, the situation remains unclear for those with HFpEF [59]. Additionally, addressing ID could enhance the effectiveness of exercise-based rehabilitation programs in HFpEF patients [61]. Managing ID may improve exercise tolerance and quality of life, but further research is needed to confirm these benefits [59,61].

High-dose IV iron, ferric carboxymaltose, or ferric derisomaltose can be administered in low-volume infusions. Ferric derisomaltose allows doses of up to 20 mg/kg in a 100 mL infusion over 30 min, enabling complete iron repletion in one session. It is recommended to routinely evaluate hemoglobin and iron status in hospitalized patients with LVEF or HFmrEF or during clinical visits when symptoms are present [28]. According to the AFFIRM-AHF and IRONMAN studies, IV iron should be administered before discharge in cases of ID in these patients [48,50].

Hemoglobin and iron levels should be monitored between 4 weeks and 4 months after the initial dose, followed by one or two annual reviews. If TSAT is below 20%, an additional dose will be necessary. Recurrent iron deficiency could indicate active blood loss. Therefore, testing for occult blood in feces and urine is suggested as a diagnostic measure [28].

The inflammatory burden, combined with ID, complicates the management of HF patients, particularly in older adults who often face concurrent conditions. This complex interplay between immunosenescence and HF-induced inflammation creates a cycle that accelerates disease progression and limits the effectiveness of therapeutic interventions, including physical rehabilitation [67]. To achieve better outcomes, clinicians should consider the impact of pharmacologic treatments on iron metabolism and tailor their management strategies accordingly.

### 5.6. Erythropoiesis-Stimulating Agents (ESAs) in HF: Clinical Considerations and Risks

Erythropoiesis-stimulating agents (ESAs), such as darbepoetin alfa and epoetin alfa, have been explored as potential therapies for anemia in HF, aiming to improve oxygen delivery and exercise capacity [68]. However, their clinical utility remains controversial due to safety concerns and limited efficacy in HF patients [69].

The RED-HF trial, one of the most extensive studies evaluating ESAs in HF, found that darbepoetin alfa effectively increased hemoglobin levels but did not reduce mortality or HF hospitalizations. Additionally, there was a higher incidence of thromboembolic events and stroke in patients receiving ESAs compared to placebo, raising concerns about their safety in this population [70].

From a clinical management perspective, ESAs stimulate erythropoiesis, increasing iron demand, which can lead to functional ID if iron stores are inadequate. This effect highlights the importance of ensuring sufficient iron availability before considering ESAs in HF patients. Moreover, these agents can worsen hypertension, increase blood viscosity, and promote thrombosis, which could offset potential benefits in cardiovascular populations [71].

Given these risks, current guidelines do not support the routine use of ESAs for anemia management in HF [4]. Instead, they recommend prioritizing IV iron supplementation, which has demonstrated excellent safety and efficacy in improving symptoms and reducing hospitalizations. Future research may help identify whether specific low-risk subgroups could benefit from ESA therapy without significant adverse effects [72].

## 6. Future Directions: Emerging Research and Therapeutic Approaches

Diagnostic criteria for ID in chronic HF patients may be inadequate and potentially misleading, as they rely too heavily on serum ferritin and TSAT thresholds. Ferritin alone may not accurately reflect functional ID, as inflammation and other non-iron-related factors common in HF can be influenced by inflammation. This imprecision may include patients unlikely to benefit from iron supplementation, excluding those who could experience significant clinical improvements. A re-evaluation of these biomarkers is needed to identify better the clinically relevant ID to HF. A refined definition could enable more targeted and effective therapeutic interventions in this population [10].

ID and inflammation play a key role in chronic HF, with distinct inflammatory pathways potentially requiring tailored therapeutic approaches. HFrEF and HFpEF may exhibit different inflammatory profiles, suggesting that a standardized approach to iron supplementation may not be optimal for all patients. This variation underscores the need for targeted research, especially in elderly patients with multiple comorbidities, who are often underrepresented in clinical trials despite constituting a significant portion of the chronic HF population [73]. Ongoing studies aim to address specific ID mechanisms across different HF phenotypes by examining the role of inflammation and its impact on iron metabolism. The ongoing IRON-PATH II (NCT05000853) seeks to elucidate the specific mechanisms of ID across different HF phenotypes by examining the role of inflammation and its impact on iron metabolism. By clarifying these mechanisms, this study hopes to pave the way for more individualized and effective iron repletion strategies that consider the unique inflammatory and iron-handling characteristics of each HF subtype [74].

A promising area of research involves targeting IL-6 to mitigate inflammation in HF patients. Ziltivekimab, a human monoclonal antibody targeting the IL-6 ligand, has effectively reduced inflammatory biomarkers in patients with chronic kidney disease and elevated hs-CRP levels [75]. The ongoing HERMES trial evaluates the effects of ziltivekimab on morbidity and mortality in patients with HFmrEF or HFpEF and systemic inflammation [76]. Tocilizumab, another IL-6 inhibitor, has been explored in inflammatory conditions such as rheumatoid arthritis and has shown potential in reducing systemic inflammation [77]. While preliminary studies suggest that IL-6 inhibition may play a role in cardiovascular diseases, its specific effects in HF remain to be fully established [78,79]. Further research is necessary to confirm the efficacy and safety of IL–6 targeted therapies in managing inflammation and ID in HF.

### Limitations and Controversies in Iron Supplementation

Although trials like FAIR-HF and CONFIRM-HF demonstrated the benefits of IV ferric carboxymaltose in improving symptoms and exercise capacity in HfmrEF, specific analyses revealed no significant differences in hard endpoints such as mortality or hospitalizations when compared to placebo. The limited sample sizes, short follow-up durations, or the inherent heterogeneity of HF populations might explain these discrepancies. Such findings highlight the need for further research to determine whether specific subgroups may derive more pronounced benefits from iron supplementation.

A deeper understanding of iron homeostasis has provided insights into the interrelation of iron biomarkers, presenting new opportunities for improving ID management in HF patients [80]. In this regard, clinical trials are testing agonists and blockers for therapeutic use, mainly testing them for use in hematological diseases [81]. In HF, treatment with an SGLT2i has shown biomarker changes consistent with improved iron utilization by increasing serum transferrin receptors and reduced ferritin, TSAT, and hepcidin. The reduction in plasma hepcidin is consistent with an enhanced capacity for iron absorption and increased mobilization of iron from sequestered stores [82]. A post hoc analysis of the IRONMAN trial revealed a trend to a more significant increase in hemoglobin with ferric derisomaltose in ID patients taking an SGLT2i at baseline, as compared with those not taking one [82].

Additionally, findings from the Randomized Trial of Empagliflozin in Nondiabetic Patients with Heart Failure and Reduced Ejection Fraction (EMPA-TROPISM trial) suggest that empagliflozin, an SGLT2i, reversed cardiac remodeling and increased physical capacity in stable nondiabetic patients with HFrEF [67]. A post hoc study explores whether treatment effects in this cohort, comprising patients with a high ID prevalence, were related to iron metabolism. The analysis indicated that empagliflozin might positively influence iron metabolism in HF patients by reducing inflammation and influencing hepcidin levels and iron homeostasis [83]. These results align with the observed hepcidin-lowering effects of SGLT2i, supporting their potential to improve iron mobilization and utilization in patients with HF with ID by enhancing the release of iron from storage sites and facilitating its availability for erythropoiesis [83]. Sub-analyses from studies such as DAPA-HF suggest a modest rise in hemoglobin and hematocrit in patients with anemia, even though this was not a primary outcome [66].

The hepcidin modulation capability by dapagliflozin, an SGLT2i, is under investigation in the ADIDAS trial (NCT04707261), a study conducted on a large cohort of anemic HF patients [84]. Further studies are needed to elucidate whether SGLT2is can contribute to restoring iron balance and could be positioned as potentially beneficial drugs and another tool for treating ID [84]. Available evidence indicates that, although most studies on correcting ID in HF use IV iron, data suggest that SGLT2i indirectly improves iron homeostasis—primarily by reducing hepcidin and thereby increasing serum iron. While no specific clinical trials are focusing on correcting ID in older adults with HF through SGLT2is, an increasing number of reports show a partial improvement of chronic (functional) anemia in patients with HF and type 2 diabetes under these medications, reinforcing a potentially beneficial interaction between SGLT2i, iron availability, and hematological parameters [82].

Exacerbated inflammation contributes to functional ID by increasing hepcidin levels, which restricts iron availability and exacerbates anemia [31]. Therefore, addressing inflammation with anti-inflammatory therapies is crucial in improving iron homeostasis in this scenario. Recently, the COLICA trial demonstrated a significant reduction in inflammation markers in acute HF patients treated with colchicine [85]. However, despite these changes in biomarkers, the study did not observe significant improvements in hard clinical endpoints, such as mortality or hospital readmission rates. These findings suggest that colchicine effectively reduces inflammation, but its broader clinical impact on HF remains uncertain. There is potential value in the further investigation of anti-inflammatory therapies specifically designed to address the unique inflammatory mechanisms at play in HF with ID. We can develop more targeted anti-inflammatory strategies that enhance iron mobilization and patient outcomes by refining our understanding of inflammation’s role in iron metabolism across different HF phenotypes, including HFrEF and HFpEF and acute versus chronic presentations [70].

Emerging therapies such as Roxadustat, a Prolyl Hydroxylase inhibitor, represent a promising future direction for addressing anemia and iron metabolism disorders in HF. Roxadustat has shown the ability to reduce hepcidin levels and enhance iron absorption, effectively improving iron availability. While its current focus lies in nephrology, its application in HF populations could open new avenues for managing anemia and functional ID, especially in patients with concurrent CKD. Future trials in HF patients will be essential to evaluate its efficacy and safety within this context [86,87].

Future directions should focus on developing specific recommendations for iron therapy in HFpEF, as there is currently a lack of evidence-based studies supporting its use in this population despite the high prevalence of ID [9]. While IV iron therapy is well-supported for patients with HFrEF, the clinical complexity of HFpEF patients, most frequently presented in older people who often present with additional comorbidities and distinct inflammatory profiles, warrants further investigation. Ongoing and future clinical trials addressing the unique characteristics of HFpEF and interaction with ID will be crucial to determine whether iron supplementation could provide meaningful clinical benefits and establish more effective, tailored treatment strategies for this population [40,59].

## 7. Conclusions

ID and chronic inflammation are critical factors exacerbating the burden of HF, especially in older adults. The interplay between inflammaging, multimorbidity, and functional ID significantly worsens HF outcomes, leading to higher rates of hospitalizations and mortality. While IV iron therapy has demonstrated benefits in improving symptoms and quality of life and reducing hospitalizations in patients with HFrEF, its role in HFpEF remains uncertain due to limited evidence.

Current diagnostic criteria for ID in HF, primarily based on serum ferritin and TSAT, may not adequately reflect iron status in chronic inflammation. Emerging data suggest prioritizing TSAT thresholds over ferritin to more accurately identify patients who could benefit from iron supplementation. Additionally, novel therapies targeting inflammation and iron homeostasis, such as SGLT2i, show promise in improving iron mobilization and HF outcomes.

Future research should focus on refining diagnostic biomarkers for ID, exploring anti-inflammatory strategies, and conducting robust clinical trials to establish evidence-based recommendations for iron therapy in HFpEF and older adults with HF. Personalized treatment approaches addressing the unique inflammatory and iron-handling profiles of HF subtypes and patient populations are essential to optimize clinical outcomes.

## Figures and Tables

**Figure 1 biomedicines-13-00462-f001:**
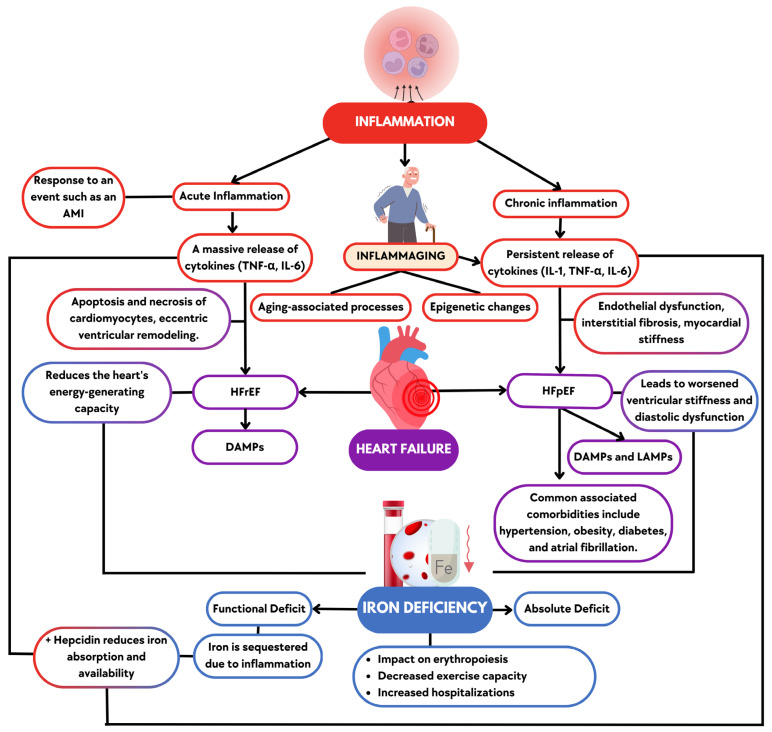
Inflammation and its role in heart failure with preserved and reduced ejection fraction. Acute inflammation, triggered by events such as heart attacks, is more closely associated with the development of heart failure with reduced ejection fraction (HFrEF), while chronic inflammation is more commonly linked to heart failure with preserved ejection fraction (HFpEF). Both types of inflammation, along with their respective forms of HF, contribute to impaired heart function and ID, which further exacerbates cardiac dysfunction. The background colors in the figure indicate specific concepts: blue represents ID, red denotes Inflammation, and purple corresponds to HF. Overlapping areas feature a combination of colors to emphasize the interplay between these conditions. AMI (acute myocardial infarction), DAMPs (damage-associated molecular patterns), HFpEF, HFrEF, IL-1 (Interleukin-1), IL-6 (Interleukin-6), LAMPs (lifestyle-associated molecular patterns), and TNF-α (Tumor Necrosis Factor-alpha). The blue represents Iron Deficiency, the red represents Inflammation, and the purple represents Heart Failure. Additionally, in the overlapping areas, the boxes are divided into two colors to highlight the relationship between these elements.

**Figure 2 biomedicines-13-00462-f002:**
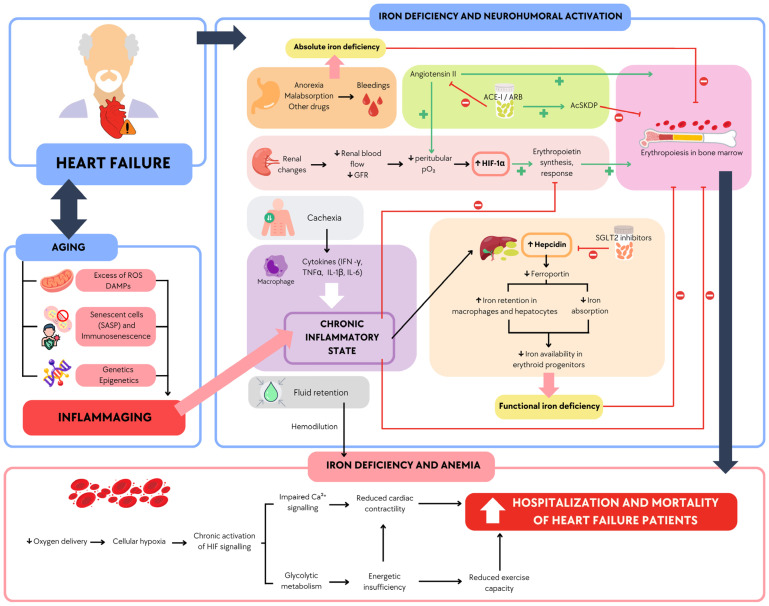
Main effects and interactions associated with HF in older patients: inflammation and iron deficiency. Older patients with HF experience renal changes, fluid retention, anorexia, poor nutrient absorption, cachexia, and immune and cytokine dysregulation, leading to a chronic inflammatory state. Additionally, due to age-related processes, they are affected by factors such as excess reactive oxygen species and cellular senescence, which contribute to systemic and chronic inflammation. Combined with medications used to treat HF, these factors result in anemia and functional iron deficiency, leading to increased mortality and hospitalizations in HF patients. Red lines indicate inhibition, while green lines represent positive stimulation. AcSDKP (N-acetyl-seryl-aspartyl-lysyl-proline), ACE-I (angiotensin-converting enzyme inhibitor), ARB (angiotensin receptor blocker), DAMPs (damage-associated molecular patterns), GFR (glomerular filtration rate), IL-1β (interleukin-1β), IL-6 (interleukin-6), ROS (reactive oxygen species), and SGLT2i (Sodium-Glucose Cotransporter-2 inhibitor).

**Table 1 biomedicines-13-00462-t001:** Differences in inflammatory mechanisms in HFrEF and HFpEF.

Aspect	HFrEF	HFpEF
Type of Inflammation	Primarily acute and chronic low-grade inflammation	Chronic low-grade inflammation
Triggering Events	AMI leading to an intense inflammatory response	Comorbidities and lifestyle factors create an unresolved inflammatory state
Primary Targets	Cardiomyocytes, leading to apoptosis and necrosis	Endothelial cells and interstitial tissue
Mechanism of Action	Proinflammatory cytokines (e.g., TNF-α and IL-6) promote inflammation and cell infiltration	Increased myocardial stiffness and fibrosis due to chronic inflammation
Outcomes of Inflammation	Eccentric remodeling, ventricular dilation, and decreased contractile function	Impaired diastolic function and increased myocardial stiffness
Associated Patterns	PAMPs and DAMPs	DAMPs with LAMPs
Influencing Factors	Focused on myocardial damage and inflammatory cell infiltration	Systemic syndrome influenced by aging, diabetes, hypertension, obesity, and atrial fibrillation

HFrEF: heart failure with reduced ejection fraction; HFpEF: heart failure with preserved ejection fraction; AMI: acute myocardial infarction; TNF-α: Tumor Necrosis Factor-alpha; IL-6: Interleukin-6; DAMPs: damage-associated molecular patterns; LAMPs: lifestyle-associated molecular patterns; PAMPs: pathogen-associated molecular patterns.

## Data Availability

No new data were created or analyzed in this study. Data sharing is not applicable to this article as it is a review of existing literature.

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
