# Peer review of "Targeting Inflammation and Iron Deficiency in Heart Failure: A Focus on Older Adults"

_biomedicines, 2025, doi:10.3390/biomedicines13020462_

Round 1
Reviewer 1 Report
Comments and Suggestions for Authors
Briefly outline the literature review approach within the Methods section.
Highlight the unique significance of addressing HF in older adults. Moreover authors should include and briefly discuss the actual HF therapy in order to enrich their discussion (doi: 10.1159/000541393.)
Include more specific insights on recent clinical trials studying SGLT2 inhibitors and their effects on ID.
Authors should explore the potential of anti-inflammatory treatments beyond colchicine, like IL-6 inhibitors.
Avoid redundancy, such as multiple mentions of IL-6’s role in hepcidin regulation.
Authors should use consistent terminology throughout: refer to "HF with reduced ejection fraction (HFrEF)" uniformly instead of alternating with "heart failure with reduced ejection fraction."
Author Response
Dear Reviewer,
Thank you very much for your valuable recommendations. We have carefully addressed each of your suggestions as follows:
- We have added a section describing the methodology used in this review.
- We have highlighted the importance of addressing HF in older adults in the introduction.
- We have included a paragraph referencing SGLT2 inhibitors and their effects on iron deficiency.
- We have incorporated information on IL-6 inhibitors, specifically ziltivekimab and tocilizumab.
- To avoid redundancy, we have streamlined the discussion of IL-6 and hepcidin, keeping the detailed pathophysiological explanations only in the initial section.
- We have reviewed the consistency of terminology throughout the manuscript and have proofread the text using two grammar-checking tools to enhance clarity and readability.
We truly appreciate your insightful feedback, which has helped us refine and strengthen our manuscript.
Best regards,
Daniela Maidana
Reviewer 2 Report
Comments and Suggestions for Authors
I have read the review article titled "Targeting Inflammation and Iron Deficiency in Heart Failure: A Focus on Older Adults" that was sent to me for evaluation. My comments, criticisms and suggestions are listed below:
First of all, I would like to congratulate the authors for this review article written on a relatively new and current topic.
1. The writing style of the article is fluent and understandable
2. There are grammatical and spelling errors here and there and should be corrected
3. Figure 2 is blurry and the writing is unreadable and should be corrected.
4. According to the new guidelines, iron supplementation should be considered for the improvement of symptoms, exercise capacity, and quality of life in patients with heart failure. However, in some large-scale publications, Among ambulatory patients who had heart failure with a reduced ejection fraction and iron deficiency, there was no apparent difference between ferric carboxymaltose and placebo with respect to the hierarchical composite of death, hospitalizations for heart failure, or 6-minute walk distance. Evaluating these articles and including them in the discussion will provide the reader with an opportunity to make an objective assessment.
5. Ferric derisomaltose has been used in some studies. It would be good if a comment is made regarding the difference between the two iron forms and their effects on the results.
6. Discussing the reasons why some studies have not shown a strong effect of IV iron treatment on symptoms or quality of life in HFpEF patients will increase the contribution of the article to the literature.
7. Discussing the use of erythropoiesis-stimulating agents in heart failure patients and their possible side effects would also be good.
Best regards
Author Response
Dear Reviewer,
Thank you very much for your recommendations. We have carefully addressed each of your suggestions as follows:
- We reviewed the terms' consistency throughout the manuscript and ran the text through two grammar-checking tools to ensure an improved and polished writing style.
- We replaced Figure 2, as the version included was an error. Thank you for pointing this out.
- We added a new subsection titled “5.2 Limitations and Controversies in Iron Supplementation”, where we discuss the findings of the HEART-FID trial and its implications.
- We discussed the differences between iron formulations and their outcomes in the section “5.1 Clinical Trials Supporting Intravenous Iron Therapy”.
- We analyzed and discussed why some studies have not shown significant differences in clinical endpoints (such as mortality or hospitalizations), particularly in “5.2 Limitations and Controversies in Iron Supplementation”.
- We introduced a new subsection titled “5.5.1 Erythropoiesis-Stimulating Agents (ESAs) in HF: Clinical Considerations and Risks”, where we mention ESAs and their side effects. Although their use is not currently recommended for anemia management in HF due to safety concerns, we acknowledged their historical importance in this context.
We appreciate your insightful feedback, which helped us improve the manuscript significantly.
Best regards,
Daniela Maidana
Round 2
Reviewer 1 Report
Comments and Suggestions for Authors
Congratulations to the authors for having improved their manuscript.